# Biobanking for discovery of novel cardiovascular biomarkers using imaging-quantified disease burden: protocol for the longitudinal, prospective, BioHEART-CT cohort study

Katharine A Kott,[1,2,3] Stephen T Vernon,[1,2,3] Thomas Hansen,[1,3] Christine Yu,[1,3] Kristen J Bubb,[1,3] Sean Coffey,[4] David Sullivan,[3,5,6] Jean Yang,[5,7] John O'Sullivan,[3,5,8] Clara Chow,[3,9,10] Sanjay Patel,[3,5,8,11] James Chong,[3,9,10] David S Celermajer,[3,8,11] Leonard Kritharides,[3,12,13] Stuart M Grieve,[3,5,8,14] Gemma A Figtree[1,2,3,5]

KAK and STV contributed equally.

For numbered affiliations see end of article.

**Correspondence to**
Gemma A Figtree;
gemma.figtree@sydney.edu.au

## ABSTRACT

**Introduction** Coronary artery disease (CAD) persists as a major cause of morbidity and mortality worldwide despite intensive identification and treatment of traditional risk factors. Data emerging over the past decade show a quarter of patients have disease in the absence of any known risk factor, and half have only one risk factor. Improvements in quantification and characterisation of coronary atherosclerosis by CT coronary angiography (CTCA) can provide quantitative measures of subclinical atherosclerosis—enhancing the power of unbiased 'omics' studies to unravel the missing biology of personal susceptibility, identify new biomarkers for early diagnosis and to suggest new targeted therapeutics.

**Methods and analysis** BioHEART-CT is a longitudinal, prospective cohort study, aiming to recruit 5000 adult patients undergoing clinically indicated CTCA. After informed consent, patient data, blood samples and CTCA imaging data are recorded. Follow-up for all patients is conducted 1 month after recruitment, and then annually for the life of the study. CTCA data provide volumetric quantification of total calcified and non-calcified plaque, which will be assessed using established and novel scoring systems. Comprehensive molecular phenotyping will be performed using state-of-the-art genomics, metabolomics, proteomics and immunophenotyping. Complex network and machine learning approaches will be applied to biological and clinical datasets to identify novel pathophysiological pathways and to prioritise new biomarkers. Discovery analysis will be performed in the first 1000 patients of BioHEART-CT, with validation analysis in the following 4000 patients. Outcome data will be used to build improved risk models for CAD.

**Ethics and dissemination** The study protocol has been approved by the human research ethics committee of North Shore Local Health District in Sydney, Australia. All findings will be published in peer-reviewed journals or at scientific conferences.

### Strengths and limitations of this study

- ► BioHEART-CT is a prospective, longitudinal cohort study assessing patients with suspected coronary artery disease (CAD) undergoing CT coronary angiography (CTCA) across multiple Australian centres.
- ► Quantitative measures of coronary atherosclerosis from CTCA datasets will be used in conjunction with biological samples, improving the power to discover new mechanisms and markers for CAD.
- ► Samples stratified by imaging-quantified disease severity will be analysed by both candidate and unbiased 'omics' approaches, using modern technologies and bioinformatic analysis to discover new biomarkers using proteomics, metabolomics, lipidomics, transciptomics, genomics and immunophenotyping.
- ► Longitudinal outcome data will be used to build risk models, which can incorporate both traditional and novel risk factors and biomarkers.
- ► Limitations of the study include selection bias, as only patients with a clinical indication for CTCA are included, and the geographical isolation of Australia, which may result in the need for confirmation of results in multinational studies.

**Trial registration number** ACTRN12618001322224.

## INTRODUCTION

Cardiovascular disease has persisted as a major cause of human morbidity and mortality despite continual improvements in preventative therapies. In 2015, cardiovascular disease accounted for one-third of all deaths worldwide,[1] and ischaemic heart

disease (IHD) remains the leading cause of years of life lost for high and middle sociodemographic groups internationally.[2] This large burden of disease puts significant stress on the health systems of individual countries. In Australia, cardiovascular disease remains the number one killer of Australians,[3] with local health costs estimated to be $A7.7 billion per annum.[4]

The standard modifiable cardiovascular risk factors (SMuRFs)[5] for atherosclerosis—hypertension, diabetes mellitus, hyperlipidaemia and smoking—were identified in epidemiological studies in the 1960s.[6] The identification and subsequent efforts to target these risk factors at community and primary care levels have led to a substantial reduction in mortality from cardiovascular disease.[1] However, substantial disease burden remains and importantly, individual variability in susceptibility to these risk factors is considerable. Conditions, such as impaired glucose tolerance and prehypertension, may also contribute to cardiovascular risk. While continuing our efforts to tackle societal and modifiable risk factors, identifying undiscovered mechanisms that lead to the development of atherosclerosis is critical. Such work will provide new biomarkers for early detection of subclinical atherosclerosis and open avenues for improved preventative and therapeutic strategies that may be relevant to those with and without known risk factors.

The importance of new detection mechanisms for subclinical atherosclerosis is highlighted by interrogating the 'fine print' of clinical studies, including information not usually presented in the main tables of manuscripts. Although data regarding the percentage of patients suffering myocardial infarction despite having no SMuRFs are often omitted, a large meta-analysis found that >50% of women and >60% of men presenting with their first acute coronary syndrome (ACS) had 0 or 1 SMuRF,[7] and similar proportions of SMuRFless patients have been identified in ACS cohort studies.[5 8] In a recent single centre Australian study, we showed that the proportion of patients presenting with ST-elevation myocardial infarction with 0 SMuRFs has risen from 11% to 27% over an 8-year period.[5] These patients have developed extensive coronary artery disease (CAD) without having any of the red flags, which would allow doctors to identify their risk of disease and intervene early. Although this is a relatively small piece of the overall IHD pie, the global burden of CAD is immense with an estimated prevalence of 470 million,[9] which makes this 25%–30% SMuRFless patient population important from a public health perspective. It is sobering to think that we have made no clear advances that allow us to identify risk and subclinical disease in this subgroup. Indeed, we also have limited information regarding how these patients respond to traditional secondary preventative strategies, and whether their long-term outcomes and disease progression differs from those with traditional modifiable risk factors.

Further evidence that novel biological mechanisms contribute substantially to atherosclerosis can be seen in results from large scale genome-wide association studies which have reported that 66% of the identified genes conveying heritable cardiovascular risk are not associated with the traditional risk factors.[10] Many of these genes were related to inflammatory processes, and studies such as the Canakinumab Anti-inflammatory Thrombosis Outcome Study (CANTOS) trial[11] confirm the importance of inflammation in CAD. However, some of these important genes have not yet been well characterised, and are not yet associated with any known pathway in atherosclerosis and CAD, highlighting the importance of ongoing discovery work.

To address this gap in our understanding of CAD pathophysiology, we have designed a unique cohort study with biobanked samples to facilitate investigation of novel cardiac risk factors and biomarkers. This biobank includes advanced, quantitative imaging measurements of coronary artery atherosclerosis to allow for accurate phenotyping of the patient groups. This overcomes a weaknesses of previous genomics studies that have relied on the presence or absence of a coronary artery event to classify individuals as diseased or as a control, improving the statistical power of the subsequent analysis. The advanced imaging dataset and clinical data, including baseline and follow-up, are integrated with the collection of a broad range of biological samples for discovery work, using state-of-the-art molecular phenotyping platforms and computational bioinformatics.

### Objectives
#### The primary objectives of the BioHEART-CT study are
1. To identify new biomarkers to assist in early CAD risk identification and stratification.
2. To identify new mechanistic pathways that may be targeted with novel therapeutic strategies to abrogate CAD risk.

#### The secondary objectives of the BioHEART-CT study include the following
1. To determine the predictive value of a modified CT coronary angiography (CTCA) scoring system for CAD risk which incorporates plaque composition data.
2. To identify new risk factors for CAD based on integration of clinical information profiles, CTCA results and clinical outcomes.
3. To develop a novel risk scoring system incorporating clinical risk factors, novel biomarkers and CTCA scores to more accurately predict CVD events.

### METHODS AND ANALYSIS
#### Patient and public involvement
The initial concept for the BioHEART-CT study was prompted by patients who had no known cardiovascular risk factors who presented with ACS and wanted to know 'why me?' The ensuing discussions with the patients and their families reinforced the public interest in research in this area. Informal consultation with patient groups through cardiovascular non-profit organisations helped build the initial framework for the study, and patient feedback

regarding the recruitment process was sought and was positive in nature. When results of interest to the general public are available, summaries will be posted to the Australia and New Zealand Clinical Trials Registry. Information about this registry is included on the patient information sheet.

## Study type

BioHEART-CT is a multicentre, prospective cohort study of patients with suspected CAD, which brings together detailed clinical information with advanced imaging and molecular phenotyping. Recruitment commenced in 2017 and is ongoing, currently with expansion into a multi-centre study.

## Study population

A total of 5000 patients will be recruited from participating hospitals and associated imaging sites. All recruitment centres are large tertiary hospitals within the city of Sydney, Australia.

### Inclusion criteria

- ▶ Patients who have been referred for CTCA for investigation of suspected CAD.
- ▶ Age 18 years or older.
- ▶ Willing and able to provide informed consent.

### Exclusion criteria

- ▶ Patients highly dependent on medical care who are unable to provide informed consent.
- ▶ Patients unwilling or unable to participate in ongoing follow-up.

## Study protocol

### Screening and enrolment

Eligible patients having a CTCA are invited to participate in the study. Eligibility is confirmed according to the inclusion and exclusion criteria, the study is explained, and informed consent obtained (figure 1).

### Patient data collection

The baseline questionnaire is completed by the research staff conducting a face-to-face patient interview. Data obtained include the following information: demographic data including smoking and alcohol intake, medical history, relevant family history, medication history, history of cardiac symptoms and indication for CTCA. Data are entered into secure, encrypted databases in a deidentified format. After recruitment, all patients are assigned a unique study number and all samples, imaging data and demographic information is identified by this number. The master list including identifiable data is securely stored on an encrypted server and is available only to senior research staff. Relevant details are made available to research staff for follow-up purposes.

Participants are contacted by phone at approximately 30 days and then annually to determine if any events have occurred. Information recorded for any potential outcome event includes the acuity, need for hospitalisation and indication for admission, hospital attended, treating physician, date of admission and details of the event. New medical or surgical diagnoses since last contact are also recorded. Participants in BioHEART-CT also consent to review of their electronic medical records and data linkage through the Centre for Health Record Linkage, which will be used for independent medical record reviews.

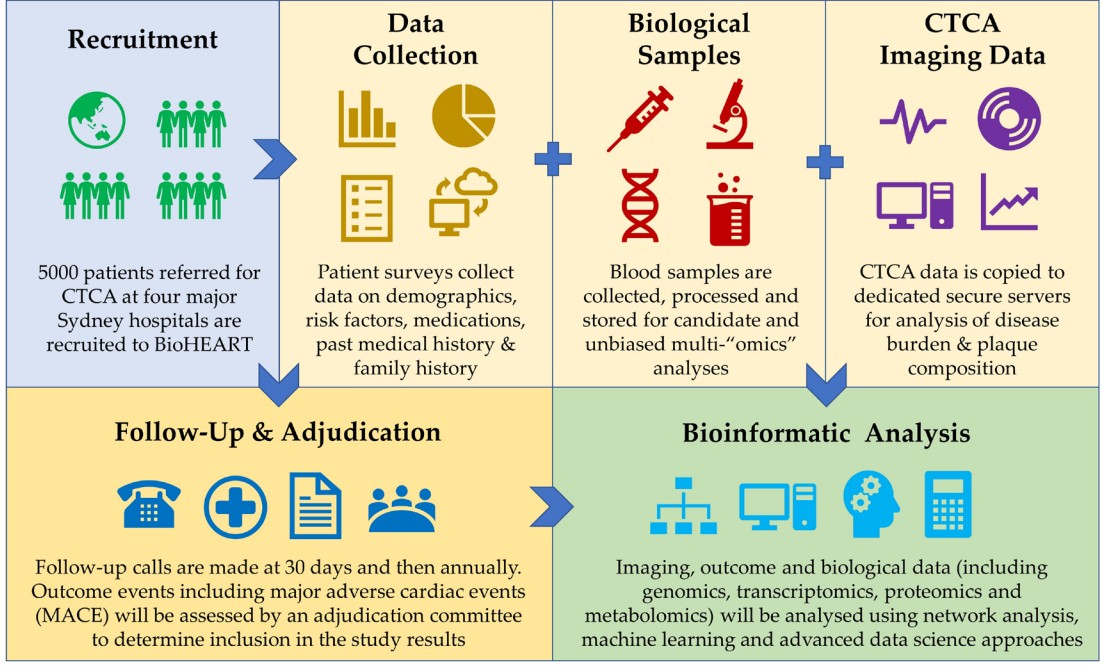

**Figure 1** BioHEART-CT study design. CTCA, CT coronary angiography.

Significant outcome events that trigger a file review to assess for major adverse cardiovascular events (MACE) or secondary outcomes are the following:

► Unstable angina requiring hospitalisation.
► Heart failure requiring hospitalisation.
► Acute myocardial infarction.
► Coronary angiogram with or without percutaneous intervention.
► Cardiac bypass graft surgery.
► Aortic or mitral valve surgery.
► Cerebrovascular accident/transient ischaemic attack.
► All-cause mortality.

All medical records will be reviewed by an independent adjudication committee to determine if an event should be included in the study results. Detailed criteria for the individual diagnoses included in the outcomes must be met.

MACE is defined as cardiovascular death, non-fatal myocardial infarction or non-fatal stroke. Exploratory outcomes include revascularisation, unstable angina or heart failure requiring hospitalisation.

### CTCA data acquisition

CTCA scans are obtained on 256 slice scanner, with reconstructions created using appropriate software for the individual machine. Each study is to be protocolled by a radiologist with at least level two accreditation from the Royal Australian and New Zealand College of Radiologists (RANZCR). All radiographers are trained in acquisition and workup of CTCA. Oral metoprolol, or ivabradine if beta blockers are contraindicated, are given to optimise the heart rate if required. Dosing is adjusted by the clinical staff according to baseline heart rate and bodyweight. Oral nitroglycerine (600–800 μg) is given immediately prior to the scan, and iodinated contrast is injected intravenously. Prospective studies are performed if the heart rate is sufficiently controlled, otherwise retrospective acquisition is used. Average dose is minimised at all participating sites in line with current recommendations.[12 13]

### CTCA imaging analysis

CT data are exported as thin digital imaging and communication in medicine (DICOM) data and are stored securely in a de-identified manner. Data are later analysed using a dedicated workstation to obtain the primary clinical scores at each site. CTCAs are analysed by standard anatomical arterial region as per the 17-segment model outlined by the Society of Cardiovascular Computed Tomography.[14] Each segment is scored according to degree of stenosis and composition of plaque (calcified, soft and mixed). The segmental scores are aggregated into a modified Gensini score (figure 2), which represents the total amount of calcified and non-calcified plaque present. Calcium scores are generated using the scanner-associated software by the Agatston method. Total raw calcium scores, raw calcium scores for each vessel, and age and sex-matched calcium percentage scores are recorded.

Further analysis will be performed at a central core laboratory using a Frontier System (syngo.via, Siemens

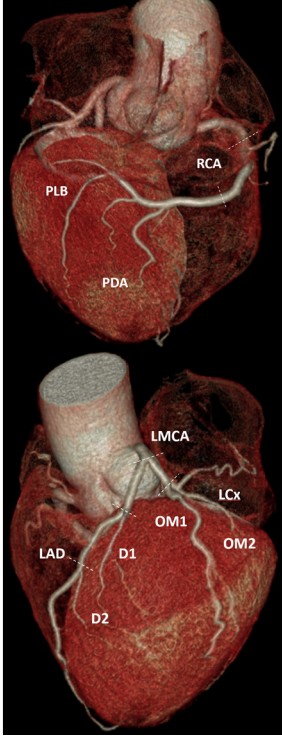

| Stenosis Location | Multiplication Factor – Right Dominant (Left Dominant) |
|---|---|
| RCA<br>- Proximal / Mid / Distal /<br>- PDA<br>- PLB | x1<br>x1<br>x0.5 |
| LMCA | x5 |
| LAD<br>- Proximal<br>- Mid<br>- Distal<br>- D1<br>- D2 | x2.5<br>x1.5<br>x1.0<br>x1.0<br>x0.5 |
| LCx<br>- Proximal<br>- Distal<br>- OM1<br>- OM2<br>- (Left PDA)<br>- (Left PLB) | x2.5 (x3.5)<br>x1 (x2)<br>x1<br>x0.5<br>(x1)<br>(x0.5) |
| Ramus | x1 |

| Percent Stenosis | Score |
|---|---|
| 0 | 0 |
| 1-25% | 1 |
| 26-50% | 2 |
| 51-75% | 4 |
| 76-90% | 8 |
| 91-99% | 16 |
| 100% | 32 |

| Plaque Composition | Multiplication Factor |
|---|---|
| Calcified | x1 |
| Mixed | x2 |
| Non-calcified (soft) | x3 |

**Figure 2** Semiquantitative plaque analysis incorporating the established Gensini scoring system[25] and adding an additional multiplier for plaque composition to create a modified Gensini score. RCA, right coronary artery; PDA, posterior descending artery; PLB, posterolateral branch; LMCA, left main coronary artery; LAD, left anterior descending artery; LCx, left circumflex artery; D1/2, diagonal branch 1/2; OM1/2, obtuse marginal branch 1/2.

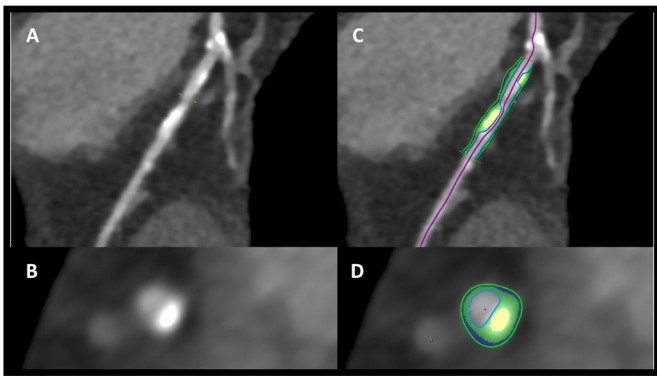

**Figure 3** Example of semiautomated analysis of a complex plaque using syngo.via running on a frontier system (Siemens healthcare, Erlangen). (A, B) show a coronary artery lesion with a large calcified component longitudinally and in cross-section. (C, D) show classification of the calcific (yellow) and fibrous (green) components of this lesion.

Healthcare, Erlangen), employing a semiautomated soft plaque algorithm (figure 3).

## Biological sample collection

A 20–30 mL blood sample is taken from an in situ intravenous cannula, or if this is not available, by venesection, prior to CTCA and drug administration. Blood is collected into ethylenediaminetetraacetic acid (EDTA), serum and citrate tubes and stored at 4°C until processing. A lithium heparin tube is collected and processed immediately to extract peripheral blood mononuclear cells (PBMCs) from the buffy coat.

Blood samples are processed by hospital pathology scientists experienced in laboratory procedures. Briefly, whole blood is set aside, then tubes are centrifuged at 1861 x g for 15 min at 4°C prior to aliquoting 500 uL samples of serum, EDTA plasma, whole blood, buffy coat and erythrocytes. The citrate tube is centrifuged at 1861 x g for a second 15 min at 4°C, after which citrate-plasma samples are aliquoted. All aliquoted samples are frozen and stored at −80°C until analysis.

For PBMC isolation, a standard gradient-separation protocol is used.[15] Briefly, blood from the lithium heparin tube is diluted 1:1 with Hanks' Balanced Salt Solution (HBSS). Diluted whole blood is then layered on top of Ficoll-Paque Plus and centrifuged (22°C, 1460 x g no brake, 20 min). The buffy coat layer containing PBMCs is then further purified with two wash steps in HBSS using repeat centrifugation as above. PBMCs are then plated at a density of ~$2.5{\times}10^4$ cells/$cm^2$ into 0.1% gelatin-coated flasks containing endothelial cell growth medium with 2% fetal bovine serum (EGM2 bulletkit, Lonza, Australia) and cultured in standard conditions for up to 21 days with regular monitoring for the spontaneous growth of endothelial progenitor cells (EPCs). Excess PBMCs are immediately frozen in bovine serum (heat-inactivated, Gibco, Australia) containing 10% dimethyl sulfoxide and stored in liquid nitrogen.

## Sample analysis

Samples will be analysed by a variety of methods with the intent of identifying candidate biomarkers and novel metabolites involved in atherosclerosis pathophysiology. There will be two principal approaches used to identify these factors. Table 1 provides a summary of all variables collected and planned assays.

**Table 1** Summary of variables and planned assays

| Clinical variables | Disease quantitation variables | Established markers | Unbiased omics approaches | Candidate approaches |
|---|---|---|---|---|
| Demographics Standard modifiable cardiovascular risk factors (SMuRFs) <br>► Hypertension <br>► Hyperlipidaemia <br>► Diabetes <br>► Smoking <br>Other medical history: <br>► BMI <br>► Family history of IHD <br>► Current medications <br>► History of cardiac symptoms | Coronary Artery Calcium Scores Gensini score Modified Gensini score (see figure 2) Frontier System semiautomated plaque analysis (syngo.via, Siemens Healthcare, Erlangen) | Cardiac <br>► Troponin <br>► NT-proBNP <br>Inflammatory <br>► CRP <br>► VCAM-1 <br>► ICAM-1 <br>► IL-6 | Metabolomics Proteomics Lipidomics Transcriptomics Genomics Immunophenotyping | Redox signalling dysregulation Endothelial cell signalling dysregulation Soluble factors released by inflammatory plaque Apoptosis signalling Mitochondrial function Angiogenesis potential Disorders of overall coagulation Soluble platelet factors Alpha-gal antibodies Heavy metal toxicities Galectin-3 |

BMI, body mass index; CRP, C reactive protein; ICAM-1, intercellular adhesion molecule 1; IHD, ischaemic heart disease; IL-6, interleukin-6; NT-proBNP, N-terminal-proB-type natiuretic peptide; VCAM-1, vascular cell adhesion molecule 1.

First is a candidate approach, investigating currently identified factors that are thought to be involved in human biological pathways that could predispose to atherosclerosis. These factors will be assessed within the cohort by specific assays that relate to the biological pathways involved. The candidate areas that have been identified for initial assessment are: soluble factors released by inflammatory signalling within the atherosclerotic plaque, dysregulated redox signalling within the endothelium, disorders of coagulation which lead to hypercoagulable states and/or hypofibrinolysis, soluble platelet factors which indicate platelet dysfunction and dysregulated signalling within endothelial cells as evidenced by assessment of cultured EPCs derived from individual patient samples. Individual assays have been designed to assess each candidate marker or pathway.

The second approach aims to investigate atherosclerosis biology in an unbiased manner by using 'omics' techniques. These assessments will use metabolomics, proteomics, transcriptomics, genomics including single nucleotide polymorphism (SNP) arrays and immunophenotyping techniques. The cells used for transcriptomics will be patient-derived EPCs and PBMCs.

Proteomics for the cohort will be assessed using a clinical proteomics platform,[16] which will give a complete description of the human peptidome in relevant samples, allowing for identification of novel or common factors worthy of further investigation. Metabolomics and lipidomics will be performed using six liquid chromatograpy-mass spectrometry platforms which will allow for identification and quantitation of a broad spectrum of metabolites, including amino acids, nucleotides, neurotransmitters and lipid subtypes.[17–21]

Immunophenotyping will be performed on PBMC samples via mass spectrometry time of flight, which allows for detailed leucocyte phenotyping on an individual basis.[22] These profiles can be associated with serum levels of established markers of inflammation such as vascular cell adhesion molecule 1 (VCAM-1), intercellular adhesion molecule 1 (ICAM-1) and interleukin-6, as well as correlated with the balance of M1 and M2 phenotype monocytes which have a well-established role in atherosclerosis.[23]

Transcriptomic and genomic assays will be used on subsets of the cohort of particular interest, who for example may display particular resistance or vulnerability to CAD based on traditional risk factor profile and disease burden. Whole transcriptome and RNA sequencing will be performed with ribosomal depletion, and sequencing of the resulting reads and counting will be based on the Genome Encyclopedia of DNA Elements. Whole genome profiling will be performed on individuals with extreme athero-susceptible or athero-resilient phenotype, allowing for unbiased discovery of genetic variants that may be causally associated with the extreme phenotypes.[24]

## Data analysis plan

Statistical machine learning approaches will be applied to these biological and clinical datasets, to identify novel pathophysiological pathways, and to prioritise new markers based on likely causal roles. Discovery analysis will be performed in the first 1000 patients of BioHEART-CT, with validation in the following 4000 patients recruited, and in international collaborators studies (figure 4). Sample size planning for developing classifiers using the high-dimensional data is based on the National Institute of Health (NIH) Biometric online calculator, assuming 1.3-fold standardised change, 50 000 molecular variables and balanced extreme groups.

Biomarker and risk factor data analysis will depend on the specifics of the assay or investigation performed. Generally, continuous variables will be presented as means (with SD) or medians (with IQRs), categorical variables as proportions (%) with $X^2$ or Fisher's exact test used to determine differences between groups. Hazard ratios for a 1 SD increase in a biomarker, after log transformation to remove effects of outliers, on MACE will be obtained from weighted Cox regression models. All p values reported will be two sided, with the 5% threshold used to determine significance.

For testing the prognostic value of prioritised biomarkers, and assuming a C-statistic of 0.75 for traditional risk factors, we estimate that we will require 705 participants to detect a C-statistic of 0.85 with p=0.05, power of 80% and an event rate of 20%. Models developed with different sets of variables and biomarkers and C-statistics for 1-year risk will be determined and compared. The model's ability to discriminate and reclassify 1-year risk will be assessed using the integrated discrimination index and the net reclassification improvement.

## DISCUSSION

Three main advances have been applied in the design of BioHEART-CT that provide a strategic advantage compared with previous studies which also used unbiased approaches to discover novel mechanisms and markers of atherosclerosis. First, this study includes quantification and characterisation of plaque (and the absence of plaque) by CTCA and advanced imaging algorithms, improving the power of the study compared with those that relied on CAD-related clinical events. Second, this study integrates high through-put, state-of-the-art multiomic molecular phenotyping with comprehensive clinical data and follow-up. And third, the study brings together a global team of biologists and bioinformaticians with complementary approaches and perspectives to enhance the likelihood of new discoveries.

In previous CAD studies, healthy controls were often defined as patients who had not had a cardiac event, or who had non-obstructive atherosclerosis on invasive coronary angiography which has limited utility in identifying extraluminal plaque. As CAD often remains silent until it is very advanced, those without a myocardial infarction could have

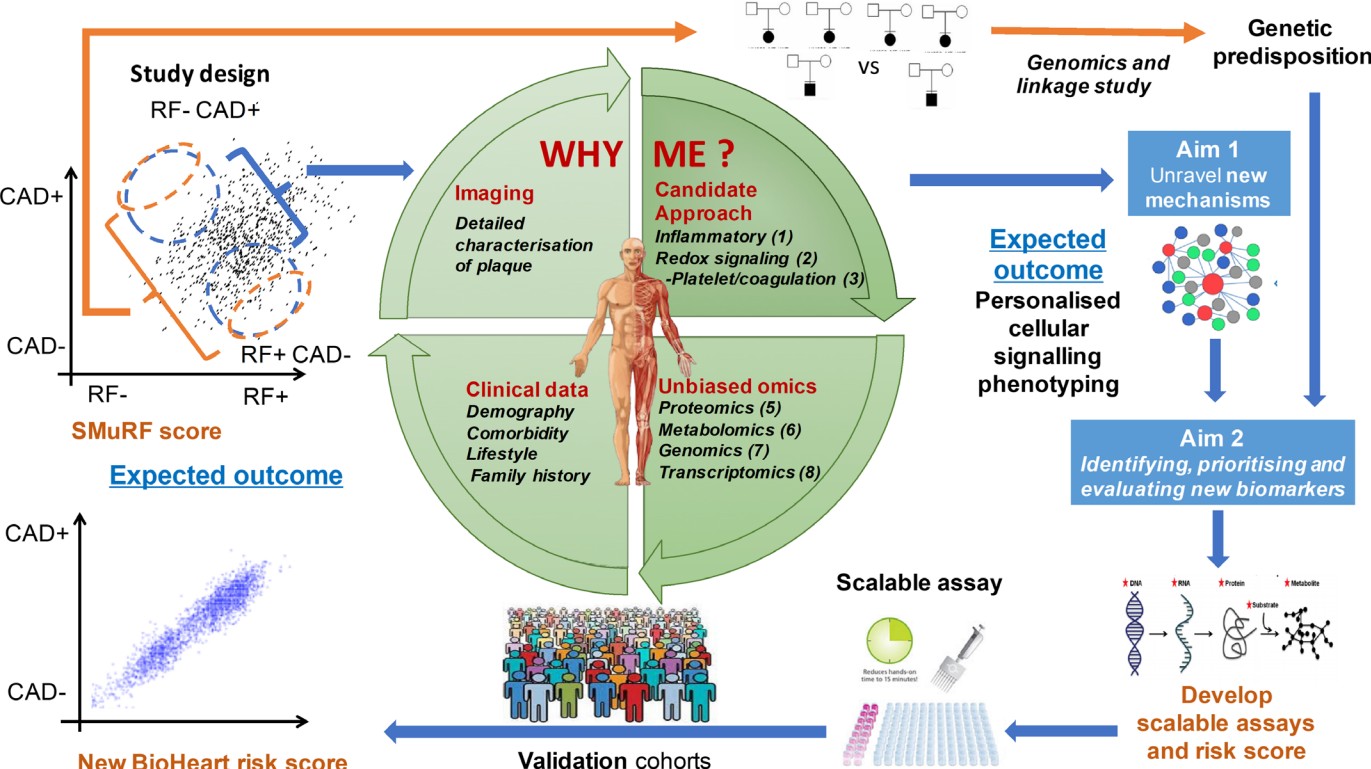

**Figure 4** Overview of the analytic approach. CAD, coronary artery disease; RF, risk factor; SMuRF, standard modifiable cardiovascular risk factor.

quite extensive disease and still be considered a 'healthy' control. In BioHEART-CT, we have the capability to use our CTCA subgroup to more precisely phenotype subjects, allowing the accurate creation of a group of healthy controls which truly have no atherosclerotic disease present in their coronary arteries. This group of healthy patients can be used as a comparator for those with various stages of disease, reducing confounding factors and enhancing the capacity to detect real differences in biology.

Additionally, CTCA phenotyping can be used to analyse plaque composition in detail. Using scoring systems such as Gensini,[25] severity scores can be generated for each individual. We have also developed a modified Gensini score as outlined above, which gives different weight to the presence of older, calcified plaque and newer, more biologically active non-calcified plaque. Automated detection of soft plaque shows promise as a scalable, reproducible method to add value to existing CTCA analysis.[26] In identifying patients with high levels of active atherosclerotic plaque, we can create groups enriched for this biological feature, with and without traditional risk factors. This will better enable us to detect novel mechanisms of CAD in those with unexplained susceptibility.

Our unbiased approach to cardiovascular risk factor discovery, applying the advantages of high throughput multiomics platforms, improved quantification of plaque and its characteristics, and advanced statistical bioinformatics, provides a high chance of discovering previously unidentified mechanisms and markers of disease. We have recently discussed the advantages of integrating

multiomics over isolated genomics studies for the complex disease process of atherosclerosis,[27] and this will be particularly beneficial in such well-phenotyped groups.

Molecular markers found to be associated with atherosclerosis may be direct drivers of disease, or result from the plaque and dysregulated vascular physiology. The closer the marker is to the underlying biology, the more likely it will be to be useful in clinical practice, transforming our ability to identify early vascular disease and susceptibility above and beyond traditional risk calculators. We are well placed to take any markers and their surrounding pathways into preclinical cellular and animal models to determine causality and/or mechanisms for association. This will help prioritise markers for testing in prospective clinical trials for their value in improving patient care and outcomes, as well as identifying potential novel signalling pathways involved in disease that may be therapeutic targets.

In conclusion, BioHEART-CT is a study created in response to patients who have had heart attacks and their frequent question from the cardiac catheter laboratory table- 'Why me?'. While it is imperative that we measure and treat well-established risk factors for atherosclerosis, this approach misses a substantial group of patients with CAD who are not identified as being at risk. We hypothesise that major mechanisms for atherosclerosis remain to be discovered and can be unravelled by comprehensive molecular characterisation, combined with powerful bioinformatic analyses. BioHEART-CT's platforms and team are well positioned for discovering new markers and mechanisms of disease, and to take these both back to the bench to unravel

new biology, as well as through to prospective clinical trial to test prognostic and clinical utility.

## ETHICS AND DISSEMINATION

Informed consent is obtained from the participants prior to enrolment, and the data and samples are deidentified and managed entirely anonymously with the exception of the required information for follow-up phone calls. The most significant risk to the patients in this study is that of venesection, with a possible consequence of mild bruising or superficial infection. Participants can withdraw from the study at any time and this will not have any impact on their clinical care.

The results of this study will be published in peer-reviewed journals and presented at domestic and international scientific meetings. No identifiable information will be published.

**Author affiliations**
[1]Cardiothoracic and Vascular Health, Kolling Institute of Medical Research, St Leonards, New South Wales, Australia
[2]Department of Cardiology, Royal North Shore Hospital, St Leonards, New South Wales, Australia
[3]Faculty of Medicine and Health, University of Sydney, Sydney, NSW, Australia
[4]School of Medicine, University of Otago, Dunedin, New Zealand
[5]Charles Perkins Centre, University of Sydney, Sydney, New South Wales, Australia
[6]Department of Biochemistry, Royal Prince Alfred Hospital, Sydney, NSW, Australia
[7]School of Mathematics and Statistics, University of Sydney, Sydney, New South Wales, Australia
[8]The Heart Research Institute, Sydney, New South Wales, Australia
[9]Westmead Applied Research Centre, Faculty of Medicine and Health, University of Sydney, Sydney, New South Wales, Australia
[10]Department of Cardiology, Westmead Hospital, Sydney, New South Wales, Australia
[11]Department of Cardiology, Royal Prince Alfred Hospital, Camperdown, New South Wales, Australia
[12]Department of Cardiology, Concord Hospital, Sydney, New South Wales, Australia
[13]ANZAC Research Institute, Sydney, NSW, Australia
[14]Department of Radiology, Royal Prince Alfred Hospital, Sydney, New South Wales, Australia

**Contributors** KK and SV participated in study design, drafted the manuscript and coordinated manuscript revisions. GF obtained the research funding and is the principal investigator of the study. All other authors (TH, CY, KB, SC, DS, JY, JO, CC, SP, JC, DS, LK and SMG) contributed to study design and revisions of the manuscript.

**Funding** GF is supported by NHMRC Practitioner Fellowship. SMG thanks the Parker Hughes Bequest for support. This study has received support from a combination of grants including from the Ramsay Teaching and Research Foundation, BioPlatforms Australia, the Vonwiller Foundation and Heart Research Australia.

**Competing interests** None declared.

**Patient consent for publication** Not required.

**Ethics approval** This study was approved by the Human Research Ethics Committee of the North Shore Local Health District, Sydney, Australia—approval number HREC/17/HAWKE/343.

**Provenance and peer review** Not commissioned; externally peer reviewed.

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
