## [Reviewer comments · BMJ Open]

ARTICLE DETAILS

TITLE (PROVISIONAL)	Biobanking for discovery of novel cardiovascular biomarkers using imaging-quantified disease burden: protocol for the longitudinal, prospective, BioHEART-CT cohort study
AUTHORS	Kott, Katharine; Vernon, Stephen; Hansen, Thomas; Yu, Christine; Bubb, Kristen; Coffey, Sean; Sullivan, David; Yang, Jean; O'Sullivan, John; Chow, Clara; Patel, Sanjay; Chong, James; Celermajer, David; Kritharides, Leonard; Grieve, Stuart; Figtree, Gemma

VERSION 1 – REVIEW

REVIEWER	Gerard Pasterkamp UMCU, the Netherlands
REVIEW RETURNED	24-Jan-2019

GENERAL COMMENTS	This paper addresses a relevant topic. The starting point is an actual issue: the changes in the baseline characteristics in the diseased population has changed over the last decades. I have some major comments regarding the high expectations that are raised but not met in the methods. - The abstract refers to machine learning, novel pathophysiological pathways and new biomarkers that will be discovered. There are many studies using -omics approaches in large cohorts in search for diagnostic algorithms for CAD. Although the CTCA procedures offer an opportunity, then the major ambition will be difficult to meet if specific -omics studies are only executed in subsets of patients if research questions pops up.- The authors refer to discovery of biomarkers driven by pathways that are involved. But how will pathways be defined and based on what data?- Established biomarkers will be measured. These should be specified.- The introduction refers correctly to the GWAS outcomes, e.g. that major part of the loci reveal genes with an unknown mechanism of action. Then in the final paragraph they mention that these GWAS studies have a weakness that will be overcome by BioHEART since now quantified measures in CTCA will be applied. Besides the fact that BioHEART will not execute genomic wide analyses I find it difficult to understand how a hard endpoint study (events in GWAS studies) can be considered a shortcoming?- Endpoint in this study are both cardiac (heart failure) as well as vascular. Is it expected that pathophysiology is similar? Are primary events (intervention based on the outcome of the CTCA) included? Then the study includes primary and secondary endpoints which may result in a mixed cohort with patients who have received more pharmacological treatment then the other.
---

	- For the -omics studies: what cell types will be studied in transcriptomics? - When will the study be initiated?
--	---

REVIEWER	G S Kanaganayagam Imperial College London, United Kingdom
REVIEW RETURNED	24-Feb-2019

GENERAL COMMENTS	The authors have clearly embarked upon a thorough investigation into the causative factors of coronary disease. The description of coronary plaque in the study appears to identify calcified, partially calcified and non calcified plaque with quantification. It is inaccurate to describe these in addition to measures of plaque vulnerability in the abstract, if these are the only measures being used. Are other known measures of vulnerability such as spotty calcification or the napkin ring sign being evaluated? There is no clear visual format of the all variables being sought in the study. Whilst the illustrations are presentable, a harder tabulated format would be useful, and give the reader a clearer view of the number of variables being studied in investigation of coronary disease. The outcomes require further definition. A novel risk score is clearly one feature that will arise from the data, which will potentially combine a variety of -omic data plus inflammatory cell data etc and correlate to coronary disease. Some factors remain unclear. The modified Gensini score needs further representation, further detail in Figure 2 may be sufficient for this - with a numeric value and weighting made more clear. What markers of inflammation are being studied? Is spontaneous growth of EPCs the only metric being used as 'assessment' of cultured EPCs.
---

REVIEWER	Ravi Dhingra University of Wisconsin Madison, WI, USA
REVIEW RETURNED	25-May-2019

GENERAL COMMENTS	BIOHEART cohort study is an initiative by Australian researchers to store blood and examine the association between serum biomarkers and subclinical atherosclerosis, specifically in patients referred for coronary CT angiography to assess coronary atherosclerosis. This study will be conducted in Australia with expected total sample size of 5000 patients. After the first 1000 patients, a discovery analysis will examine the association of biomarkers and coronary CT results whereas the remaining 4000 patients will make up a validation cohort. Because of the investigators intent to bio bank serum and use proteomics, metabolomics, lipidomics, transcriptomics, genomics and immunophenotyping to discover new biomarkers for coronary artery disease prediction, and lack of a specific hypothesis in this study protocol, a sample size calculation based on effect sizes is difficult to conduct. Nonetheless, this study intends to explore newer genomics and proteomics biomarkers and pathways which are not known so far and lead to development of subclinical atherosclerosis. An event rate of 20% is expected from these highly selected patients who
--

	already perhaps have an indication to undergo CT angiogram to assess coronary artery disease. Investigators propose to conduct this study because there is rising incidence of coronary events in patients with no prior history of traditional cardiovascular risk factors. Some prior investigators though have argued that 90% of cardiovascular atherosclerotic events do occur in patients with at least 1 or more subclinical cardiovascular risk factors (blood sugars 100-126mg/dl, or prehypertension etc). Perhaps that should be addressed in introduction as well. Data collection includes demographics, clinical as well as cardiovascular disease history and imaging studies that consist of CT scan results only. It is unclear if information on other imaging modalities such as ECG, echocardiogram or coronary angiograms (if performed) will also be collected. Follow up data is collected at 30 days and annually only by phone although detailed information regarding hospitalizations are collected through the nationally linked electronic medical records. It is unclear if outpatient data from primary care physicians can be linked, as some of these patients may not always make it to hospitalization. Additionally, if patients die at home, information regarding the cause of death can be explored by autopsy data. Perhaps the investigators may be able to discuss with patients and obtain the autopsy consent in an event if patients suddenly die at home. It is a strength of this study that an independent board reviews all hospitalization data and CT scans are performed and interpreted in a protocol-based fashion across different hospitals. One major limitation of the study is that only those patients who are referred for coronary CTA are included in this cohort and therefore has limited generalizability to the general population. Nonetheless, restricting the group to CTA cohort perhaps increases the chances for more outcomes during follow up hence shorter study duration.
--	---

VERSION 1 – AUTHOR RESPONSE

Reviewer 1

The abstract refers to machine learning, novel pathophysiological pathways and new biomarkers that will be discovered. There are many studies using -omics approaches in large cohorts in search for diagnostic algorithms for CAD. Although the CTCA procedures offer an opportunity, then the major ambition will be difficult to meet if specific -omics studies are only executed in subsets of patients if research questions pops up.

We thank the reviewer for their comment and would like to clarify that the majority of the -omics techniques will be performed on the discovery cohort of 1000, including SNP array assessment of >880,000 SNPs (PNDA array), with validation in the following 4000 patients. The more focused subset analysis utilizing transcriptomic and genomic assays referred to on page 9 will

be performed to address questions of biological interest for hypothesis generation in addition to the unbiased -omics approaches being applied more broadly. The risk models and diagnostic algorithms developed will incorporate the results from the unbiased -omics techniques applied to the discovery cohort.

The authors refer to discovery of biomarkers driven by pathways that are involved. But how will pathways be defined and based on what data?

Assessment of candidate biomarkers will be guided by the current body of scientific literature and current understanding of biochemical pathways plausibly involved in the pathophysiology of atherosclerosis. The pathways being researched will continually be evolving as new discoveries are made and published in the scientific literature. Our current planned investigations of pathways are listed in the newly created Table 1, which tabulates all the collected variables and planned assays.

Established biomarkers will be measured. These should be specified.

Thank you for identifying this omission. The text has now been updated on page 9 to specify the established biomarkers that will be measured. In addition, these biomarkers are also listed in Table 1.

The introduction refers correctly to the GWAS outcomes, e.g. that major part of the loci reveal genes with an unknown mechanism of action. Then in the final paragraph they mention that these GWAS studies have a weakness that will be overcome by BioHEART since now quantified measures in CTCA will be applied. Besides the fact that BioHEART will not execute genomic wide analyses I find it difficult to understand how a hard endpoint study (events in GWAS studies) can be considered a shortcoming?

Thank you for your comment. We would like to clarify our thinking on this point. Genomic studies will be performed on all patients to provide SNP data. The SNP data will predominantly be acquired to be able to integrate it with multiple layers of molecular and -omics data, but will also be used in conjunction with disease burden as quantified on CT. In terms of the benefit of the BioHEART approach, we acknowledge that hard endpoints are ultimately key, but it is widely accepted that a large proportion of “healthy” controls in GWAS studies may have moderate to extensive atherosclerosis without ever having had an event. In our opinion this contamination of the “healthy” control group confounds accurate biomarker assessment that could correlate with early atherosclerotic plaque formation. We feel our CT confirmation of a truly healthy subgroup is a significant advantage for biological discovery. Additionally, new biomarkers discovered using the association with plaque volume and characteristics determined by CTCA will ultimately be tested for prognostic ability against hard endpoints in both our own validation cohort, as well as external cohorts where possible.

Endpoint in this study are both cardiac (heart failure) as well as vascular. Is it expected that pathophysiology is similar? Are primary events (intervention based on the outcome of the CTCA) included? Then the study includes primary and secondary endpoints which may result in a mixed cohort with patients who have received more pharmacological treatment than the other.

As stated on page 6-7, major adverse cardiovascular events (MACE) “is defined as cardiovascular death, non-fatal myocardial infarction or non-fatal stroke. Exploratory outcomes include revascularisation, unstable angina or heart failure requiring hospitalization”. The primary events following a CTCA would be included in the MACE. The inclusion of heart failure as a secondary outcome is intended to identify subjects who developed an ischaemic cardiomyopathy requiring hospital-level intervention. We are aware of the potential heterogeneity of the pathophysiology of heart failure as a possible limitation of the study. Identification of the aetiology of the heart failure and the variability of pharmacological management will be reviewed by the adjudication committee, and this will be factored into relevant analyses as required.

For the -omics studies: what cell types will be studied in transcriptomics?

This will be performed on both patient-derived EPCs and PBMCs. The text has been updated to reflect this on page 9.

When will the study be initiated?

The study commenced in 2017, initially as a single centre study. It is in the phase of expansion into a multi-centre trial. The text on page 5 has been updated to explain this.

Reviewer: 2

The authors have clearly embarked upon a thorough investigation into the causative factors of coronary disease. The description of coronary plaque in the study appears to identify calcified, partially calcified and non-calcified plaque with quantification. It is inaccurate to describe these in addition to measures of plaque vulnerability in the abstract, if these are the only measures being used. Are other known measures of vulnerability such as spotty calcification or the napkin ring sign being evaluated?

Thank you for pointing out the discrepancy between the abstract and the text regarding measures of plaque vulnerability. While our intention is to train our automated analysis program to detect these measures, they will not be included in our initial analysis of the CTCAs. The reference to plaque vulnerability measures has been removed from the abstract.

There is no clear visual format of the all variables being sought in the study. Whilst the illustrations are presentable, a harder tabulated format would be useful, and give the reader a clearer view of the number of variables being studied in investigation of coronary disease.

Thank you for the suggestion. We have now included an additional table (Table 1) to summarize the variables and the currently planned assays.

The outcomes require further definition. A novel risk score is clearly one feature that will arise from the data, which will potentially combine a variety of -omic data plus inflammatory cell data etc and correlate to coronary disease. Some factors remain unclear. The modified Gensini score needs further representation, further detail in Figure 2 may be sufficient for this - with a numeric value and weighting made more clear.

We agree a graphical representation of the modified score would be clearer. We have addressed this by including an additional figure (new Figure 2) in in the imaging analysis section.

What markers of inflammation are being studied?

As discussed on page 9 of the manuscript, markers of inflammation will include C-reactive protein, VCAM-1, ICAM-1 and IL-6. Immunophenotyping will also give extensive information regarding the inflammatory state of patients, with detailed data available relating to the number of inflammatory cells present in the peripheral blood. This information is now summarized in Table 1.

Is spontaneous growth of EPCs the only metric being used as 'assessment' of cultured EPCs?

The EPCs will undergo a panel of cell-signaling and molecular phenotyping testing, including unbiased transcriptomics, proteomics and metabolomics. Candidate pathways for EPCs will include redox signaling, assessment of angiogenesis potential, assessment of apoptosis signaling and mitochondrial function testing. These details have been included in Table 1.

Reviewer: 3

Investigators propose to conduct this study because there is rising incidence of coronary events in patients with no prior history of traditional cardiovascular risk factors. Some prior investigators though have argued that 90% of cardiovascular atherosclerotic events do occur in patients with at least 1 or more subclinical cardiovascular risk factors (blood sugars 100-126mg/dl, or prehypertension etc). Perhaps that should be addressed in introduction as well.

Thank you for this suggestion. Whilst we acknowledge that pre-diabetes and pre-hypertension may be contributing to the population of SMuRFless ACS patients, we envisage that even in such individuals there may be identifiable markers of early atherosclerosis which may be more reliable than the more labile measures of blood sugar and blood pressure. We have amended the introduction to address pre-clinical risk factors on page 4. "Conditions such as impaired glucose tolerance and pre-hypertension may also contribute to cardiovascular risk. While continuing our efforts to tackle societal and modifiable risk factors, identifying undiscovered mechanisms that lead to the development of atherosclerosis is critical. Such work will provide new biomarkers for early detection of subclinical atherosclerosis and open avenues for improved preventative and therapeutic strategies that may be relevant to those with and without known risk factors."

Data collection includes demographics, clinical as well as cardiovascular disease history and imaging studies that consist of CT scan results only. It is unclear if information on other imaging modalities such as ECG, echocardiogram or coronary angiograms (if performed) will also be collected.

The occurrence of a coronary angiogram will be included in the outcome data, however imaging results from the procedure will not be analysed. CTCA is the only imaging modality directly utilized in this study. ECGs are not available and are not formally assessed, though a history of atrial fibrillation is collected. We have added Table 1 to give a summary of all variables being assessed, which gives a clear overview of what is being collected.

Follow up data is collected at 30 days and annually only by phone although detailed information regarding hospitalizations are collected through the nationally linked electronic medical records. It is unclear if outpatient data from primary care physicians can be linked, as some of these patients may not always make it to hospitalization. Additionally, if patients die at home, information regarding the cause of death can be explored by autopsy data. Perhaps the investigators may be able to discuss with patients and obtain the autopsy consent in an event if patients suddenly die at home.

While we will be reviewing the Centre for Health Record Linkage (CHeReL) database and the electronic medical records of any patient lost to follow-up to investigate possible adverse outcomes, we acknowledge that patients could die at home without a known cause of death. Unfortunately, the suggestion to arrange autopsy of these patients is not feasible within the limits of the Australian public health system.

It is a strength of this study that an independent board reviews all hospitalization data and CT scans are performed and interpreted in a protocol-based fashion across different hospitals. One major limitation of the study is that only those patients who are referred for coronary CTA are included in this cohort and therefore has limited generalizability to the general population. Nonetheless, restricting the group to CTA cohort perhaps increases the chances for more outcomes during follow up hence shorter study duration.

We acknowledge this as a limitation of the study which we feel is unavoidable. We believe the radiation exposure from a CTCA, while small, limits the ability to generalize this to the whole population. Part of the purpose of this study is to determine ways to risk stratify patients using serological tests which do not carry radiation risk. We agree that the restriction to clinically indicated CTCA likely increases the chance of outcomes occurring during the study duration.

VERSION 2 – REVIEW

REVIEWER	Gerard Pasterkamp UMCU The Netherlands
REVIEW RETURNED	13-Jul-2019
GENERAL COMMENTS	The authors adequately responded to my comments. I doubt whether the expectations that are depicted will be met in the presented study design. But future will tell.